# Systematic Review of Cognitive-Behavioural Therapy for Social Anxiety Disorder in Psychosis

**DOI:** 10.3390/brainsci7050045

**Published:** 2017-04-25

**Authors:** Maria Michail, Max Birchwood, Lynda Tait

**Affiliations:** 1School of Health Sciences, University of Nottingham, Jubilee Campus, Nottingham NG7 2TU, UK; Lynda.Tait@nottingham.ac.uk; 2Warwick Medical School, University of Warwick, Coventry CV4 7AL, UK; M.J.Birchwood@warwick.ac.uk

**Keywords:** social anxiety, psychosis, cognitive-behavioural therapy

## Abstract

Background: Social anxiety is highly prevalent among people with psychosis and linked with significant social disability and poorer prognosis. Although cognitive-behavioural therapy (CBT) has shown to be effective for the treatment of social anxiety in non-psychotic populations, there is a lack of evidence on the clinical effectiveness of CBT for the treatment of social anxiety when this is co-morbid in psychosis. Methods: A systematic review to summarise and critically appraise the literature on the effectiveness of CBT interventions for the treatment of social anxiety in psychosis. Results: Two studies were included in the review assessing the effectiveness of group CBT for social anxiety in schizophrenia, both of poor methodological quality. Preliminary findings suggest that group-based CBT is effective in treating symptoms of social anxiety, depression and associated distress in people with schizophrenia. Conclusion: The evidence-base is not robust enough to provide clear implications for practice about the effectiveness of CBT for the treatment of social anxiety in psychosis. Future research should focus on methodologically rigorous randomised controlled trials with embedded process evaluation to assess the effectiveness of CBT interventions in targeting symptoms of social anxiety in psychosis and identify mechanisms of change.

## 1. Introduction

Social anxiety is among the most commonly reported anxiety disorders with a 12-month prevalence of 2% reported in European countries [1] and 7.1% in the US [2]. It usually develops during childhood or adolescence, follows a chronic course [3,4,5], and has a lower likelihood of a full remission compared to other anxiety disorders [6]. Social anxiety disorder is highly co-morbid and poses a significant risk for the emergence of other anxiety and mood disorders [7]. Social anxiety is particularly prevalent in psychosis, with rates ranging between 8% and 36% [8,9,10,11,12,13,14,15,16]. Its presence in psychosis is linked with poorer prognosis, greater likelihood of an early relapse [17], and significant levels of social disability [13]. 

Cognitive-behavioural therapy (CBT) is recommended by the National Institute for Health and Care Excellence (NICE) as an indispensable treatment option for people with psychosis and as a first line treatment for those refusing antipsychotic medication [18]. Traditionally, CBT has focused on the reduction of psychotic symptoms rather than on affective co-morbidities such as depression, social anxiety and distress despite the highly debilitating nature and impact of these co-morbidities [12,13,19]. Although CBT has shown to be effective for the treatment of social anxiety disorder in non-psychotic populations [20], there is lack of evidence on the clinical effectiveness and cost-effectiveness of CBT for the treatment of social anxiety when this is co-morbid in psychosis.

We carried out a systematic review to summarise and critically analyse the evidence on the effectiveness of CBT interventions in improving social anxiety symptoms, general anxiety, distress, depression, positive and negative symptoms of schizophrenia, and quality of life in people with psychosis.

## 2. Methods

### 2.1. Study Design

This paper comprises a systematic review reported according to the Preferred Reporting Items for Systematic Reviews and Meta-Analyses (PRISMA) statement [21] following the Cochrane Handbook for Systematic Reviews of Interventions [22]. 

### 2.2. Search Strategy

We searched the following databases: Cochrane Central Register of Controlled Trials, CINAHL (Cumulative Index to Nursing and Allied Health Literature), EMBASE, MEDLINE, PsychINFO, SCI (Science Citation Index); and trial registries https://clinicaltrials.gov/ and ISRCTN for grey literature. We set the following search limits: (1) Study design: randomised controlled trials (RCTs) and quasi-experimental studies; (2) English language only. No date restrictions were applied. The electronic search strategy terms used were: (phobic disorders OR social *anxi* OR social anxiety disorder) AND (psychotic disorders OR schizophrenia) AND (Cognitive Therapy OR Cognitive Behaviour Therapy) AND (clinical trial OR cross-over studies OR double-blind method OR random allocation OR randomised controlled trial OR single-blind method). EndNote was used to record titles, abstracts and inclusion/exclusion decisions. 

### 2.3. Selection Criteria

#### 2.3.1. Types of Studies

Randomised controlled trials (RCTs) and quasi-experimental studies (pre- and post-test design) were eligible for inclusion.

#### 2.3.2. Types of Intervention

Cognitive-behavioural/cognitive interventions (group-based and one-to-one) targeting social anxiety in people with psychosis. No limitations in terms of psychological theory informing the intervention, the person delivering the intervention or the setting in which the intervention was delivered were imposed.

#### 2.3.3. Comparator

Any other treatment; no treatment; treatment-as-usual and a waiting list control were included as control conditions.

### 2.4. Types of Participants

Participants aged 16–65 years; with schizophrenia or related psychoses and social anxiety disorder, diagnosed using any recognised diagnostic criteria e.g., ICD-10 [23] or DSM-V [24] were included. Studies with a primary diagnosis of organic brain disorder were excluded. 

### 2.5. Primary Outcome

Social anxiety was assessed using any psychometrically validated scale, including self-report and clinician administered. 

### 2.6. Secondary Outcomes

Secondary outcomes included general anxiety symptoms; distress; depression; positive and negative symptoms of schizophrenia; quality of life assessed using any psychometrically validated scale, both self-report and clinician administered; cost of CBT intervention. 

### 2.7. Selection Procedure, Data Extraction and Data Management

MM and LT independently screened the title and abstract of retrieved studies for inclusion. The EPOC (Effective Practice and Organisation of Care) data extraction form and the EPOC data checklist were used to extract data from eligible studies. The researchers extracted the following data: setting, population and demographic characteristics of participants; baseline characteristics; details of intervention and control conditions; methodology; recruitment, completion and attrition rates; outcomes and times of measurement; suggested mechanisms of intervention action; information for assessment of the risk of bias. MM and LT independently assessed study quality and risk of bias using the Cochrane’s Collaboration tool for assessing risk of bias [25]. 

## 3. Data Synthesis

Due to the low number of included studies, we undertook a narrative synthesis following guidance by Popay et al. [26]. The narrative synthesis involved describing, organising, exploring and interpreting the study findings, taking into account the methodological adequacy. We investigated the similarities and differences between study findings including study design; quality; study power; intervention characteristics and delivery; participants and outcome measures. Where particular patterns of findings have emerged, we have presented possible explanations for these findings. 

## 4. Results

### 4.1. Description of Studies 

Figure 1 presents a detailed flow diagram of the study selection process. Two studies were included in the review (Table 1), both conducted in Australia. Halperin et al. [27] assessed the efficacy of a group-based CBT for social anxiety in schizophrenia using a randomly assigned design (group-based CBT vs. waitlist control) with pre-, post-, and six-week follow-up ratings. Kingsep et al. [28] investigated the effectiveness of a CBT treatment model as an intervention for social anxiety in people with schizophrenia using a controlled clinical trial (CBT vs. waitlist control) with alternation as an allocation method and a two-month follow-up. 

### 4.2. Participants

In total, 49 participants (36 males) were included in the two studies; 23 in the intervention group and 26 in the control group. Detailed demographic data were only provided in one study [27], where the mean age of the total sample (*n* = 20) was 39.6 years (range 19–67); all participants were single and only one was employed. Data on ethnicity were not provided. Participants in the two studies had a diagnosis of schizophrenia and co-morbid social anxiety determined either by a score of >20 on the Brief Social Phobia Scale or by a diagnostic structured interview. Participants in the two studies were attending a community-based living skills rehabilitation programme at the Royal Perth Hospital. Baseline mean severity levels of social anxiety (SIAS: 46.85) and depression (CDSS: 9.98) in the intervention group (*n* = 23) were not different to those in the control group (*n* = 26) (SIAS: 42.82; CDSS: 8.75). However, baseline levels of social phobia in the intervention group (BSPS: 47.21) were significantly different (*p* < 0.05) compared to those in the control group (BSPS: 38.63).

### 4.3. Interventions

The intervention in the two studies took place in a community setting (Inner City Mental Health Service of Royal Perth Hospital) and was delivered in small groups. Only one study [28] provided details of the type of therapists delivering the intervention. In the study by Kingsep et al. [28], therapy was delivered by two facilitators; a clinical psychologist and a psychiatric nurse or occupational therapist. Both intervention models were based on an evidence-based protocol for the treatment of social anxiety in the general population [29] tailored to the needs of individuals with schizophrenia. In the study by Kingsep et al. [28], the intervention consisted of twelve 2-h weekly sessions followed by a follow-up session two months after the last treatment session. In the study by Halperin et al. [27], the duration of the intervention was eight 2-h weekly sessions. The two studies reported the core intervention components which included: psycho-education, exposure, cognitive restructuring, role-play and intra-session assignments (homework). 

### 4.4. Comparisons

A waitlist control group was included in the two studies. Participants in this group received treatment as usual including community case management and antipsychotic medication. The waitlist control group received the intervention when the experimental group completed treatment. 

### 4.5. Outcomes

Outcome measures used in the two studies to assess social anxiety were the Social Interaction Anxiety Scale [30] (SIAS) and the Brief Social Phobia Scale [31] (BSPS). In addition, the Brief Fear of Negative Evaluation [32] (BFNE) was used in the study by Kingsep et al. [28]. Depression was assessed using the Calgary Depression Rating Scale [33] (CDSS). Psychological symptom patterns were assessed using the Brief Symptom Inventory [34] (BSI) and quality of life was measured with the Quality of Life, Enjoyment and Satisfaction Questionnaire [35] (Q-LES-Q). In addition, Halperin et al. [27] assessed alcohol use using the Alcohol Use Disorders Identification Test [36] (AUDIT). Three assessment points were included in the two studies: pre-treatment, post-treatment; two months follow-up in Kingsep et al. [28] and six-weeks follow-up in Halperin et al. [27]. None of the studies assessed the cost-effectiveness of the intervention. 

### 4.6. Methodological Quality

Table 2 presents the authors’ judgements regarding the methodological quality of the included studies. The main methodological problems in the two studies were lack of allocation concealment; lack of intention-to-treat-analysis and the adoption of a waitlist control design. Kingsep et al. [28] additionally suffered from biased allocation to interventions as allocation was by alternation. Blinded assessment of outcomes was not clear in the study by Halperin et al. [27]. In addition, both studies had very small sample sizes which weakened their statistical power. Fidelity was assessed only in the work of Kingsep et al. [28]; yet no fidelity data were actually presented. Finally, participants were not followed for a period of time adequate to allow a meaningful assessment of the effectiveness of the interventions. 

### 4.7. Summary of Findings

Due to the low number of included studies and the risk of bias identified in these studies, we did not perform a meta-analysis. Instead, we provide a summary of the results of each study (Table 3). In the study by Kingsep et al. [28], the authors investigated whether group CBT was effective in treating co-morbid social anxiety in individuals with schizophrenia (CBT vs. waitlist control). Assessments of social anxiety, depression, general psychological distress and quality of life were taken at pre-treatment, post-treatment and two months post-treatment. At post-treatment, the authors reported statistically significant differences between the CBT and the waitlist control group, with the former reporting significant reduction in severity levels of social anxiety (SIAS; BSPS, BFNE), depression (CDSS) and psychological distress (GSI). Participants in the CBT group also showed a significant improvement in quality of life compared to those in the control group. In terms of strength of association, large effect sizes were reported for the BFNE (d = 1.05) and CDSS (d = 1.82); medium effect sizes for the SIAS (d = 0.69) and GSI (d = 0.76) and a small effect size for the BSPS (d = 0.17). Quality of life (QLESQ) was a small to medium effect size (d = 0.42). Treatment effect in the CBT group was maintained at two months post-treatment, where levels of social anxiety and depression were significantly reduced between pre-treatment and follow-up (Table 3). Improvement in quality of life was also maintained at two months post-treatment.

In the study by Halperin et al. [27], the authors investigated the efficacy of group-based CBT in treating symptoms of social anxiety in people with schizophrenia (CBT vs. waitlist control). Measures of social anxiety (SIAS, BSPS), depression (CDSS), general psychological distress (GSI), quality of life (QLESQ) and alcohol use (AUDIT) were taken pre- and post-treatment and at six weeks post-treatment. The authors compared post-treatment change scores between the CBT and control group (Table 3). Severity levels of social anxiety, depression and general psychological distress were significantly reduced in the CBT group compared to the control group. Quality of life showed significant improvement in the CBT group compared to the control group, whereas no differences in levels of alcohol use were reported between the two groups. To assess the strength of association, we calculated effect sizes using Smith and Glass’s delta procedures [37] for those outcome measures used in both included studies, and used Cohen’s [38] definition of effect sizes as *small*, d = 0.2; *medium*, d = 0.5; and, *large* d = 0.8. Contrary to Kingsep et al. [28], a small effect size was reported for the SIAS (d = 0.3) and GSI (d = 0.13). In line with Kingsep et al. [28], CDSS was large in effect size (d = 1.76) and BSPS was small in effect size (d = 0.09). QLESQ (d = 0.38) was also of a small effect size. The authors reported that the treatment effects in the CBT group were maintained at six weeks follow-up; however, the data for this were not provided. 

## 5. Discussion

This is the first systematic review of the evidence on the effectiveness of CBT intervention for the treatment of social anxiety disorder in people with psychosis. Two studies were included in the review, both conducted in Australia. The intervention models in the two studies were based on an evidence-based protocol for the treatment of social anxiety in the general population tailored to the needs of individuals with schizophrenia. The two studies adopted a group-based format for the delivery of CBT and a waitlist control group. 

Serious methodological problems were reported in the two studies, including lack of allocation concealment; lack of intention-to-treat-analysis and the adoption of a waitlist control design. In addition, small sample sizes weakened the statistical power in both studies. The fidelity of interventions (i.e., the extent to which the intervention was implemented as planned) was either not assessed [27] or relevant information was not presented [28]. 

Due to the low number of included studies in this review and their poor methodological quality, it was not possible to conduct a meta-analysis. A description of the results of each study was presented instead. In both studies the effectiveness of the CBT intervention was demonstrated with participants in the CBT group reporting significantly lower severity levels of social anxiety at post-treatment compared to those in the control group. These effects were maintained at two months follow-up in the study by Kingsep et al. [28] and at six weeks follow-up in the study by Halperin et al. [27]; although the latter did not present the relevant data to support this. Depression scores as well as symptoms of general psychopathology in the CBT group in the two studies significantly improved at post-treatment, and these effects were maintained at follow-up. In addition, an increase in quality of life scores was reported among CBT group participants in both studies. Therefore, preliminary findings suggest that group-based CBT is effective in treating symptoms of social anxiety, depression and associated distress in people with schizophrenia. However, significant methodological flaws mean that overall these studies do not provide a robust evidence-base on which to reach firm conclusions about the effectiveness of CBT for the treatment of social anxiety when this is co-morbid in psychosis. 

## 6. Implications for Practice and Research

CBT, recommended by NICE [39] for people with psychosis, has traditionally focused on reducing psychotic symptoms, in particular hallucinations and delusions, and not co-morbid depression and social anxiety [40]. There is a knowledge gap in relation to the effectiveness of CBT in treating affective dysregulation, particularly social anxiety when this is co-morbid in psychosis. This systematic review aimed to address this knowledge gap; however, the evidence we have identified, appraised and synthesised is not robust enough to provide clear implications for practice. Some tentative suggestions which can be made are: group-based CBT seems to be effective in treating symptoms of social anxiety, depression and distress in people with psychosis. The intervention model was based on the protocol formed by Heimberg et al. [29] for the treatment of social anxiety in the general population tailored to the needs of individuals with schizophrenia. Core intervention components included psycho-education, exposure, cognitive restructuring, role-play and intra-session assignments (homework).

Treatment studies using methodologically sound RCTs to evaluate the effectiveness of interventions aimed at treating affective dysregulation and particularly social anxiety in people with psychosis is scarce. Given the elevated prevalence and debilitating nature of social anxiety when co-morbid in psychosis, there is a pressing need for methodologically rigorous studies in evaluating complex CBT interventions for the treatment of social anxiety disorder with an emphasis on identifying mechanisms of change. There is also need to adapt and modify existing CBT models for social anxiety to address the specific nature of symptoms and difficulties experienced by people with psychosis [41]. Our previous work [40] suggests that “conventional” CBT models for social anxiety in psychosis could be considerably enhanced with an additional focus on shame and entrapment cognitions linked to psychosis and accompanying concealment behaviours which form part of the safety behaviour repertoire of socially anxious psychotic individuals. Our research into the psychological underpinnings of social anxiety in psychosis shows that dysfunctional appraisals held by socially anxious psychotic people are characterised by shamefulness, humiliation and perceived rejection by others [40]. We have argued that awareness of the social stigma attached to a diagnosis of psychosis is internalised and endorsed by people with psychosis who accept their “schizophrenic” identity. This internalised stigma subsequently leads to increased shamefulness and fear of the illness being revealed to others due to the consequences of this discovery (e.g., social exclusion, marginalization). Hence, people with psychosis will attempt to conceal their stigmatised identity by engaging in safety behaviours e.g., avoidance, withdrawal from social interactions, as a way of minimising the anticipated threat i.e., discovery, and ‘saving’ the individual from the consequences of such a social threat, for example being shamed, humiliated and rejected by others. We have argued, however, that the use of safety behaviours could be counterproductive as it can contaminate social interactions by promoting behaviours of submissiveness, avoidance and withdrawal in people with psychosis. An RCT with an embedded process evaluation focusing on shameful cognitions, alongside perceptions of entrapment and reducing or eliminating concealment linked behaviours as potential mechanisms of change could be effective in psychosis. To do this, we recommend a three-arm trial including CBT for social anxiety based on traditional models such as Heimberg [29] vs. modified CBT for social anxiety with a focus on shame, entrapment, internalized stigma and perceived social rejection as mediators of change vs. a control group. We highlight the importance of qualitative research methods such as interviews or focus groups as essential for identifying how the modified CBT works, for whom and why, as well as any variations in the delivery of the intervention which could affect outcomes, e.g., distress, anxiety, and depression.

## 7. Limitations

The language of the included studies was restricted to English due to feasibility issues and lack of resources. The low number and poor quality of the included studies is an additional limitation of this review, particularly with relation to implications for practice.

## 8. Conclusions

This systematic review provided a narrative synthesis of the evidence on the effectiveness of CBT interventions for the treatment of social anxiety disorder when this is co-morbid in psychosis. Although preliminary evidence shows that group-based CBT tailored to the needs of people with schizophrenia could be effective in treating co-morbid symptoms of social anxiety, depression and distress, serious methodological flaws in the included studies make it difficult to reach firm conclusions. Future research should focus on methodologically rigorous randomised controlled trials with embedded process evaluation assessing the effectiveness of CBT interventions while identifying mechanisms of change to inform how the intervention works and why, thus facilitating future implementation. 

## Figures and Tables

**Figure 1 brainsci-07-00045-f001:**
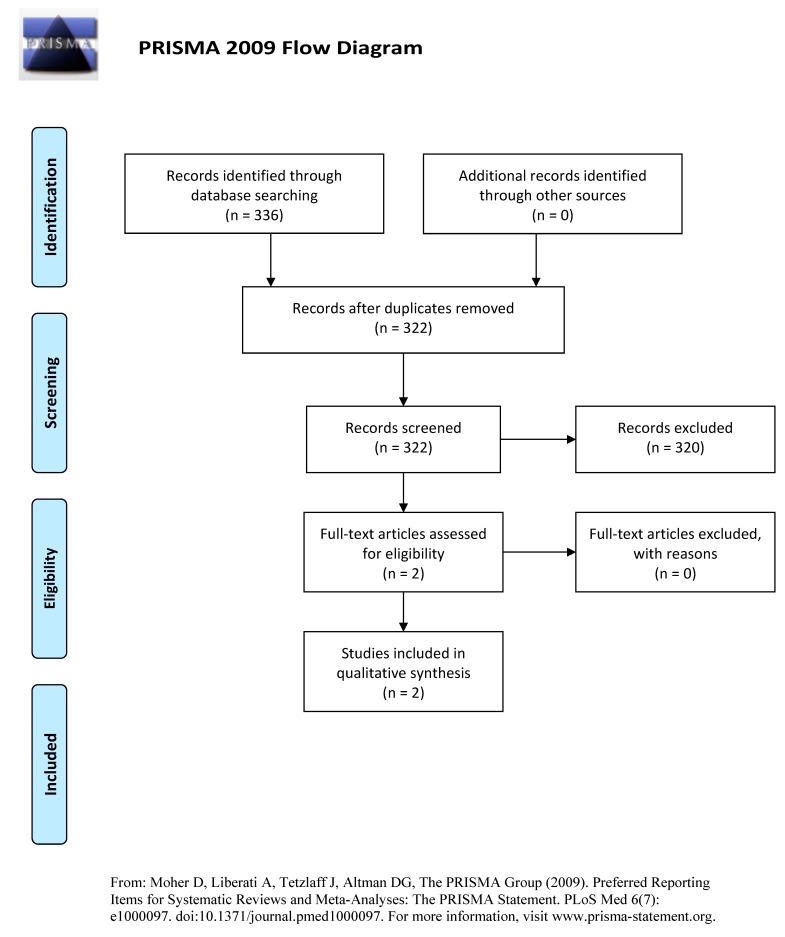
PRISMA flow diagram.

**Table 1 brainsci-07-00045-t001:** Overview of characteristics of included studies.

Study	Country	Design	Participants	Intervention	Control	Outcome Measures
Halperin et al. (2000) [27]	Australia	Randomised controlled trial (no details of randomisation provided)	16 (13 males)	Group CBT based on Heimberg et al. (1995) [28] model; eight 2-h weekly sessions	Waitlist-control group	Social anxiety: BSPS; SIAS Depression: CDSS Quality of life: QLESQ Psychological symptom: BSI Alcohol use: AUDIT
Kingsep et al. (2003) [28]	Australia	Controlled trial with alternation as allocation method	33 (no data on gender)	Group CBT based on Heimberg et al. (1995) [28] model; twelve 2-h weekly sessions plus one follow-up session	Waitlist-control group	Social anxiety: BSPS; SIAS; BFNE Depression: CDSS Quality of life: QLESQ Psychological symptom: BSI

CBT: Cognitive-behavioural therapy; BSPS: Brief Social Phobia Scale; SIAS: Social Interaction Anxiety Scale; CDSS: Calgary Depression Rating Scale; BSI-GSI: Brief Symptom Inventory-Global Severity Index; QLESQ: Quality of Life, Enjoyment and Satisfaction Questionnaire; AUDIT: Alcohol Use Disorders Identification Test; BFNE: Brief Fear of Negative Evaluation.

**Table 2 brainsci-07-00045-t002:** Risk of bias assessment of included studies.

	Halperin et al. (2000) [27]		Kingsep et al. (2003)[28]	
	Judgement	Support for Judgement	Judgement	Support for Judgement
Random sequence generation (selection bias)	Low risk	Quote: “participants were randomly allocated”	High risk	Allocation by alternation
Allocation concealment (selection bias)	High risk	Quote: “if a participant assigned to the treatment group could not participate due to another commitment; he was thus assigned to the control group”	High risk	Not done—allocation was by alternation
Blinding of participants and personnel (performance bias)	Low risk	Participants and therapists could not be blind	Low risk	Participants and therapists could not be blind
Blinding of outcome assessment (detection bias) (patient-reported outcomes)	Unclear risk		Low risk	Assessors were independent from the therapists and blind to the patients’ treatment conditions
Incomplete outcome data addressed (attrition bias)	High risk	Control group: 2 participants excluded from analysis (1 moved away; 1 was hospitalised). Intervention group: 2 participants withdrew during week 2	High risk	41 initially consented and a total of 8 dropped out of the study—all were in the intervention group and dropped out at session 2 or 3
Selective reporting (reporting bias)	Low risk	All outcome measures reported	Low risk	All outcome measures reported
Other bias	High risk	Waitlist control received the intervention	High risk	Waitlist control received the intervention. More than one outcome assessor—no data on inter-rater reliability

**Table 3 brainsci-07-00045-t003:** Results of included studies.

Study	Pre-Treatment (Mean; SD)		Post-Treatment (Mean; SD)		
	Intervention	Control	Intervention	Control	*p* Value
Halperin et al. (2000) [27]					
Social anxiety					
BSPS	47.29 (10.63)	37.56 (13.58)	38.14 (6.23)	37 (13.18)	*
SIAS	45.14 (11.26)	41.11 (12.61)	37.43 (11.89)	40.88 (11.39)	*
Depression					
CDSS	10.71 (2.43)	8.56 (3.50)	4.57 (3.26)	9.33 (2.70)	**
Psychological distress					
BSI–GSI	71.86 (5.73)	64 (6.12)	64.86 (10.59)	64.11 (5.75)	**
Quality of life					
QLESQ	52.22 (11.85)	54.79 (12.35)	58.75 (10.65)	54.50 (11.32)	**
Alcohol use					
AUDIT	11.29 (9.14)	6.67 (8.83)	8.43 (5.68)	7.11 (9.24)	ns
Kingsep et al. (2003)[28]					
Social anxiety					
BSPS	47.13 (11.79)	39.71 (12.16)	36.81 (7.12)	38.82 (11.66)	**
SIAS	48.56 (10.01)	44.53 (15.03)	34.44 (11.25)	44.24 (14.25)	**
BFNE	49.78 (10.12)	46.88 (8.43)	37.78 (11.98)	48 (9.75)	**
Depression					
CDSS	9.25 (3.36)	8.94 (3.83)	4.06 (2.89)	9.29 (2.87)	**
Psychological distress					
BSI–GSI	53.38 (24.30)	57.25 (17.19)	46.38 (20.97)	57.74 (15.01)	*
Quality of life					
QLESQ	49.43 (12.80)	54.65 (11.83)	59.03 (8.64)	54.23 (11.44)	**

* *p* < 0.05; ** *p* < 0.01; BSPS: Brief Social Phobia Scale; SIAS: Social Interaction Anxiety Scale; CDSS: Calgary Depression Rating Scale; BSI-GSI: Brief Symptom Inventory-Global Severity Index; QLESQ: Quality of Life, Enjoyment and Satisfaction Questionnaire; AUDIT: Alcohol Use Disorders Identification Test; BFNE: Brief Fear of Negative Evaluation.

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
