# Peer review of "Systematic Review of Cognitive-Behavioural Therapy for Social Anxiety Disorder in Psychosis"

_brainsci, 2017, doi:10.3390/brainsci7050045_

Round 1

Reviewer 1 Report

Summary of aims & contribution of paper: The paper highlights that individuals diagnosed with psychosis frequently report difficulties with anxiety in social situations; however, there is a lack of studies that evaluate whether cognitive behaviour therapy can reduce symptoms of social anxiety when co-morbid with a diagnosis of psychosis. The study aimed to provide a systematic review of the literature. Due to the small number of studies that met the inclusion criteria (n=2), a narrative synthesis was adopted with a conclusion that further RCTs are required. The main contribution of the paper is for future studies to evaluate whether focusing on shame, humiliation, stigma and rejection in therapy can reduce symptoms of social anxiety for individuals also diagnosed with psychosis.

Introduction: The study is an interesting and potentially useful area of focus. With the movement towards transdiagnostic cognitive and behavioural approaches, I was rather surprised to see a dated focus on diagnoses as discrete ‘disorders’ with the view that they require a disorder-focused ‘treatment’ approach, instead of acknowledging the complexity of how distress is experienced and consequently the range of symptoms observed. I would like to see some acknowledgement in the introduction of the transdiagnostic approach to working with psychological distress and why the authors opted to take the approach they have to reviewing the literature (i.e., justification) –this is particularly important due to the focus on co-morbidity. Researchers and clinicians cannot ignore the figures; research, both national and international has shown rates of co-morbidity to be high, with approximately half of all psychological disorders being co-morbid (Farrell et al., 2001; Goodwin et al., 2003; Kessler et al., 2005; Robins et al., 1991; Ustan et al., 1995) and 32-80% of individuals diagnosed with one disorder will be diagnosed with a co-morbid disorder during their lifetime, with a sixth of the population diagnosed with multiple concurrent disorders (Kessler et al., 2005).  My question to the authors is whether it is useful to continue to view and treat distress through a disorder-focused lens? There is an emerging evidence base for transdiagnostic approaches, Dr Sara Tai at the University of Manchester, for example, is currently using Method of Levels Therapy (MOL), which is a transdiagnostic cognitive therapy with clients diagnosed with psychosis.    

Data from the studies: There is no critical evaluation of the pre data scores between groups for the two studies. The authors state that the baseline mean severity levels were not different between the intervention and control group (line 126). Looking at the scores, I can see that SIAS is 46.85 for the intervention group and SIAS: 38.63 for the control group. This is a concrete statement without statistical tests being performed – the authors are advised to word this more carefully. This is particularly important as in table 3, scores for the BSPS are 47.29 (pre intervention) and 38.14 (post intervention), which are similar in difference to the SIAS and were found to be statistically significant. Please provide a critical review of pre scores between the two groups.

Discussion: The section on ‘Implications for practice and research’ is the strongest part of the paper with discussions around shame, humiliation, stigma and rejection, with an acknowledgement that therapy should focus on these areas – this could be given more of a platform within the paper.

Recommendation for future RCTs as the answer: There are limitations to RCTs and the conclusions that can be drawn from them, regardless of sitting at the upper end of the hierarchy of evidence, which are not acknowledged here. In order to add anything useful to the literature, we as researchers need to continue to conduct rigorous research but that is clinically meaningful. RCTs do not actually tell us much about the mechanisms of change. I would like to see a more detailed outline for future research rather than the generic statement that more methodologically rigorous RCTs are needed – what would that involve?

Consider reviewing:

-          Bonell, C., Fletcher, A, Morton, M., Lorenc, T. & Moore, L. (2012).  Realist randomised controlled trials: A new approach to evaluating complex public health interventions. Social Science and Medicine, 75, 2299-2306.

-          Cartwright, N. (2010). What are randomised controlled trials good for? Philosophical Studies, 147, 59-70.

-          Elliott, R. (2010). Psychotherapy change process research: Realizing the promise. Psychotherapy Research, 20, 123-135.   

Author Response

Comment 1: Introduction: The study is an interesting and potentially useful area of focus. With the movement towards transdiagnostic cognitive and behavioural approaches, I was rather surprised to see a dated focus on diagnoses as discrete ‘disorders’ with the view that they require a disorder-focused ‘treatment’ approach, instead of acknowledging the complexity of how distress is experienced and consequently the range of symptoms observed. I would like to see some acknowledgement in the introduction of the transdiagnostic approach to working with psychological distress and why the authors opted to take the approach they have to reviewing the literature (i.e., justification) –this is particularly important due to the focus on co-morbidity. Researchers and clinicians cannot ignore the figures; research, both national and international has shown rates of co-morbidity to be high, with approximately half of all psychological disorders being co-morbid (Farrell et al., 2001; Goodwin et al., 2003; Kessler et al., 2005; Robins et al., 1991; Ustan et al., 1995) and 32-80% of individuals diagnosed with one disorder will be diagnosed with a co-morbid disorder during their lifetime, with a sixth of the population diagnosed with multiple concurrent disorders (Kessler et al., 2005).  My question to the authors is whether it is useful to continue to view and treat distress through a disorder-focused lens? There is an emerging evidence base for transdiagnostic approaches, Dr Sara Tai at the University of Manchester, for example, is currently using Method of Levels Therapy (MOL), which is a transdiagnostic cognitive therapy with clients diagnosed with psychosis.  

Reply: We thank the reviewer for his/her feedback on the introduction. We do not agree that we have adopted a disease-focused approach to CBT. On the contrary, we have acknowledged that social anxiety and associated distress are highly co-morbid and that social anxiety poses a significant risk for the emergence of other anxiety and mood disorders (lines 36-37). Furthermore, we have emphasized the problem with traditional CBT approaches which have focused on the reduction of psychotic symptoms rather than on affective co-morbidities such as depression, social anxiety and distress despite the highly debilitating nature and impact of these co-morbidities (lines 43-46). To this end, we propose that the focus of future CBT models in psychosis should focus on affective dysregulation (anxiety, distress, depression) and underlying appraisals (shame, internalised stigma, perceived social rejection) rather on reducing psychotic symptoms (Discussion). In fact, recently published work by the authors (2014) highlights the effectiveness of their CBT intervention in reducing harmful compliance to commanding voices and associated distress in a group of people with command hallucinations, without a change in positive psychotic symptoms including hallucinations.

Comment 2: Data from the studies: There is no critical evaluation of the pre data scores between groups for the two studies. The authors state that the baseline mean severity levels were not different between the intervention and control group (line 126). Looking at the scores, I can see that SIAS is 46.85 for the intervention group and SIAS: 38.63 for the control group. This is a concrete statement without statistical tests being performed – the authors are advised to word this more carefully. This is particularly important as in table 3, scores for the BSPS are 47.29 (pre intervention) and 38.14 (post intervention), which are similar in difference to the SIAS and were found to be statistically significant. Please provide a critical review of pre scores between the two groups.

Reply: We thank the reviewer for this comment. We apologise that due to a typographical error the wrong mean values for the SIAS and BSPS were provided.  We provide the correct values below and have amended the relevant section in the paper accordingly: “Baseline mean severity levels of social anxiety (SIAS: 46.85) and depression (CDSS: 9.98) in the intervention group (n=23) were not different to those in the control group (n=26) (SIAS: 42.82; CDSS: 8.75). However, baseline levels of social phobia in the intervention group (BSPS: 47.21) were significantly different (p<0.05) compared to those in the control group (BSPS: 38.63).”

Comment 3: Recommendation for future RCTs as the answer: There are limitations to RCTs and the conclusions that can be drawn from them, regardless of sitting at the upper end of the hierarchy of evidence, which are not acknowledged here. In order to add anything useful to the literature, we as researchers need to continue to conduct rigorous research but that is clinically meaningful. RCTs do not actually tell us much about the mechanisms of change. I would like to see a more detailed outline for future research rather than the generic statement that more methodologically rigorous RCTs are needed – what would that involve?

Reply: We thank the reviewer for his/her suggestions and references provided. We agree that there are limitations to RCTs and perhaps a more realist evaluation of CBT within the context of an RCT will reveal important information about mechanisms of change i.e. which intervention components are ‘active ingredients’; who does this interventions work for and why, facilitating, thus, later implementation. We have amended our discussion and conclusions accordingly, as well as the abstract of the study.

References

Birchwood M., Michail M., Meaden A., Tarrier N., Lewis S., Wykes T., Davies L., Dunn G., Peters E. (2014). Cognitive behaviour therapy to prevent harmful compliance with command hallucinations (COMMAND): a randomised controlled trial. Lancet Psychiatry, 1, 23-33

Reviewer 2 Report

Manuscript Review: Systematic review of cognitive-behavioural therapy for social anxiety disorder in psychosis

Authors: Maria Michail, Max Birchwood, Lynda Tait

Reviewer: Virginia V W McIntosh

Overall

The Manuscript is a systematic review of studies examining the effectiveness of social anxiety treatments for individuals with psychosis.

The review is important, as social anxiety is more common in individuals with psychosis and is highly impactful on functioning.

The manuscript has not previously been published, and there are no conflicts of interest in this review. 

Abstract

The Abstract is an accurate summary of the manuscript, and is able to be understood as a stand-alone summary. There are no discrepancies between the Abstract and the remainder of the manuscript. The conclusions could be strengthened and made more informative by the addition of a statement about the implications for practice as well as for future research.

Introduction

The Introduction is concise, with appropriate referencing to key research articles in the area. The review’s purpose is clearly defined, and a strong rationale for performing the study is based in the review of existing research and clinical need.

Methods

The Methods of the review are clearly outlined, which enables the reader to follow the search strategy and methods of analysis of research included in the review. It would be feasible for another researcher to reproduce the study. Choices of strategy are justified. The decision to provide a narrative synthesis of the two available studies in the review, rather than to conduct a meta-analysis is described. The methods as described are appropriate to meet the stated aims of the study.

Results

Results are clearly explained, and explanation/analysis are provided. The order of presentation is appropriate, and guides the reader through the search process, and analysis and interpretation of the research. Although the results review only two studies, the exploration and interpretation of those studies’ findings, and the clear paucity of existing research is an important result. No results are included in this section that have not been preceded by an appropriate description in the Methods section.

Discussion

The Discussion is concise and centres around two themes. The first relates to the paucity of existing research on this important topic, the quality and results from the two studies reviewed, and the need for further research. This is a timely and succinct review. The discussion of the small sample sizes of the two studies and the implications for statistical power, while accurate, are less important than some of the other methodological issues. In the two studies, in spite of the small sample sizes, differences between the treatment and control groups were found. Were no differences found, the issue of statistical power and sample size would have been more relevant, as it is unknown whether no differences are due to the absence of effectiveness or the intervention or insufficient power to detect differences. The authors tentatively conclude that there is weak evidence that CBT appears effective in treating social anxiety in individuals with psychosis. The second theme is around important clinical insights in the area and how these might be incorporated into future research and the clinical practice of CBT for social anxiety in psychosis that future research will study.

Limitations of the study are noted.

Tables and Figure

The tables are well laid out, and appropriately summarise and describe the important results from the review. Table 3 would benefit from a key to the abbreviations for the psychometric scales from the two studies. All tables and The PRISMA flow diagram would benefit from more descriptive titles so that they can be understood without referring to the remainder of the manuscript.

References

The reference list largely follows the format for the journal. One exception is that the full author list is not included for some references. My understanding is that the style for the journal uses all authors, or up to ten authors followed by et al. Also no full stops appear at the end of each reference.

The reference list is appropriate for the review, and the authors appear to have appropriately represented salient points from the sources referenced. 

Kessler et al (line 289) does not have a year of publication.

Minor issues

The word ‘data’ is treated as singular and used with a singular verb form. This should be changed to reflect data as plural.

When the two studies are referred to together, the authors use ‘both studies’, which reads as if the studies are connected, although they are not. I would suggest using ‘the two studies’ to avoid this inadvertent implication.

A duplicate reference format appears at line 163.

Halperin is misspelt at line 222

Summary Opinion

In summary, the manuscript is a well-conducted and clearly written review article. It reports a systematic review of research on cognitive-behavioural therapy for social anxiety disorder in individuals with psychosis.

Author Response

We thank this reviewer for their constructive feedback and comments.

Below see below how we addressed their comments.

Add statement about implications for practice and research in the abstract.

We have now added in the abstract an appropriate statement with relation to the study's implications about practice.

2. Add abbreviations and full titles for scales in Table 3.

We have now added the full titles and abbreviations in Table 3.

3. The titles of the tables and PRISMA should be more descriptive.

Where necessary and relevant, we have refined the titles we used in our figures and tables.

4. Add full stop after each reference.

We have now added a full stop after each reference in the reference list.

5. Full references for up to 10 authors.

We have now added all author names for those references with up to 10 authors.

6. Kessler et al (line 289) publication year is missing.

Publication year for this reference has now been added.

7. Use plural for data.

We have now corrected this and all reference to the word data is in plural.

8. Instead of using "both studies" use instead "in the two studies".

Where appropriate, we have changed our reference to "both studies" to "in the two studies".

9. Line 163 duplicate reference format.

We have now corrected this typo.

10. Line 222 Halperin misspelt.

We have now corrected this typo.

Round 2

Reviewer 1 Report

Line 29 – should read ‘mechanisms’.

Line 249 – typographical issue ‘so social’.

Line 292-3 – awkwardly constructed sentence  ‘…to inform how the intervention works and why, facilitating, thus future implementation’.

The reviewer holds a very different position to the authors in relation to what a disease-focused approach entails, and so has not asked for further changes.